# Trophic Positions of Sympatric Copepods across the Subpolar Front of the East Sea during Spring: A Stable Isotope Approach

Dong-Hoon Im [1] and Hae-Lip Suh [2,*]

1   Marine Environment Research Division, National Institute of Fisheries Science,
    Busan 46083, Republic of Korea
2   Department of Oceanography, Chonnam National University, Gwangju 61186, Republic of Korea
*   Correspondence: suhhl@jnu.ac.kr

**Abstract:** We investigated the trophic relationship between particulate organic matter (POM) and sympatric copepods within the epipelagic zone (~200 m depth) in the East Sea during spring based on stable isotope analysis (SIA). The SIA indicated that interspecific differences in the prey size and vertical segregation of feeding migration range among copepods may promote niche partitioning among sympatric copepods in each region of the subpolar front (SPF). Additionally, our results showed remarkable differences in the copepod community structure and resource utilization across the SPF. The south region of the East Sea showed higher species richness of copepods than the north region, while copepods that fed mainly on POM in the surface and subsurface chlorophyll maximum layers showed smaller body sizes in the south region. These results revealed that the food chain between primary producers and higher trophic levels was longer in the south region than in the north region. Additionally, $\delta^{13}C$ and $\delta^{15}N$ values of copepods increased gradually with the body size increase whereas $\delta^{15}N$ values in the north region showed the reverse trend. Latter results could be attributed to the consumption of deep-layer POM in small copepods. Therefore, we suggest that northward shifts in the distribution of copepods under global warming may decrease energy efficiency in the pelagic ecosystem of the East Sea.

**Keywords:** resource partitioning; East Sea; copepod community structure; energy flow; subpolar front; climate change





## 1. Introduction

The East Sea is a semi-enclosed marginal sea surrounded by the Republic of Korea, Japan, and Russia connected to the northwestern Pacific through narrow and shallow straits, including Korea/Tsushima, Tsugaru, Soya, and Tatar Straits. Oceanic features, such as independent thermohaline circulation, deep-water formation, a subpolar front (SPF), and mesoscale eddies similar to those of the North Atlantic Ocean, make the East Sea frequently cited as a "miniature ocean" for investigating large-scale oceanic processes [1,2]. Recently, the East Sea has been experiencing notable environmental changes including a water mass structure and climate regime shift [3,4], which accompany changes in the species composition of plankton and fish, and ecosystem processes.

Copepods are a major component of marine zooplankton and play an important role in carbon cycling and zooplankton population dynamics in the ocean [5,6]. Thus, they not only play a role in the trophic linkage between primary producers and higher trophic levels but also act as opportunistic omnivores that utilize various types of particulate organic matter (POM) within the water column [7,8]. In addition, the distribution of copepod species is strongly influenced by hydrographic conditions, such as water temperature and salinity [9,10], and has been considered a biological indicator of water masses [11]. Coexistence is possible when sympatric species show differences in ecological requirements and have different life strategies, food preferences, or a distributional overlap [12]. However,

difficult access to species-specific characteristics, such as resource utilization and feeding habitats among copepods, make the effects of the copepod community structure on energy flow in pelagic ecosystems not fully understood [13,14].

In the East Sea, the SPF is present around 38–40 °N and forms a boundary between a warm-water mass, the East Korea Warm Current, and a cold-water mass, the southward-flowing Liman Current and the North Korea Cold Current (Figure 1), thus separating the temperate and sub-arctic copepod communities [15,16]. With global warming, the acceleration of volume and heat transport through the Tsushima Warm Current can cause changes in zooplankton distribution patterns and species interactions, which have the potential to influence carbon flow in the pelagic ecosystem [17,18]. Moreover, ectotherms shift the distribution pattern of organisms to where they can achieve optimal performance within their thermal limit [19,20]. Jung [21] found that a water temperature increase within the epipelagic zone in the East Sea could increase the population of warm-water species, including anchovy, club mackerel, and common squid, while sardines, which are relatively cold-water species, would nearly disappear in the south region. Joo et al. [22] reported that annual primary production in the entire region of the East Sea showed a decreasing trend over the last decade. Therefore, the decrease in primary production and northward shifts in the longer food chain ecosystem under global warming may influence future fishery productivity in the East Sea.

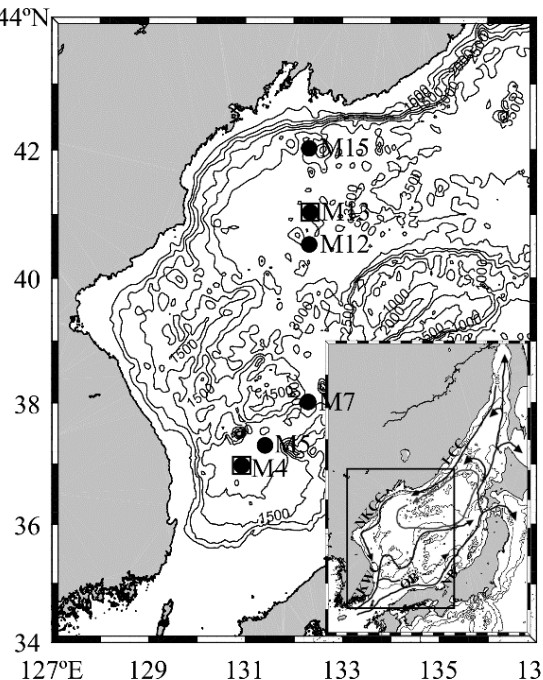

**Figure 1.** A sampling station map in the East Sea. CTD, fluorescence, particulate organic matter, and zooplankton data were obtained at six stations (●), and stable carbon and nitrogen isotope ratios of zooplankton were obtained at two stations (□). Inset shows the general location of the study area, and isobaths are shown at 500 m intervals. The major currents in the East Sea are from Park et al. (2013). EKWC, East Korea Warm Current; LCC, Liman Cold Current; NB, Near-Shore Branch of Tsushima Warm Current; NKCC, North Korea Cold Current; OB, Off-Shore Branch of Tsushima Warm Current.

Stable carbon and nitrogen isotope ratios of consumers provide time-integrated information on resource utilization and environmental inputs and have been extensively used as a natural tracer for the foraging strategies of migrating species in various ecosystems [23,24]. In the pelagic ecosystem, the isotopic gradient of suspended POM reflects the vertical distribution of nutrient sources and processes in relation to biological incorporation or geochemical cycling [25,26]. The stable carbon and nitrogen isotope ratios

of zooplankton undergoing vertical migration can be influenced by the vertical profiles of POM with regard to the feeding habitat [26,27]. Recent studies have shown that the size-based approach of pico- (<2 μm), nano- (2–20 μm), and micro-POM (20–200 μm) can provide essential information on vertical feeding migration and habitats of copepods within the water column [28,29]. However, they focus on a few filter-feeding zooplankton and, thus, provide limited information on the effect of zooplankton community structure on energy flow in the pelagic ecosystem [13,14].

In this study, we measured carbon and nitrogen stable isotopic ratios of copepods and pico- to micro-POM at the surface (0 m, 10 m), subsurface chlorophyll maximum (SCM), 100 m, and 200 m depth layers in the south and north regions of the SPF in the East Sea. To evaluate differences in the trophic position of sympatric copepods in each region, we compared the $\delta^{13}C$ and $\delta^{15}N$ values of copepods with pico- to micro-POM in each region.

## 2. Materials and Methods

### 2.1. Samplings

Hydrographic data, water and zooplankton samples were obtained at six stations in the East Sea by the R/V Akademik M.A. Lavrentyev in April 2014 (Figure 1). Temperature, salinity, and fluorescence data were obtained with a CTD (SBE 911Plus, Seabird Electronics Inc., Bellevue, WA, USA). Water samples were also obtained at 0 m, 10 m, SCM, 100 m, and 200 m depth layers using a CTD-rosette system with 20-L Niskin bottles. For pico- to micro-POM sampling, water samples (~5 L) obtained at each depth were pre-filtered through 200- and 20-μm sieve in succession, and micro-POM samples were collected on pre-washed and pre-combusted (450 °C for 4 h) 25 mm Whatman GF/F filters (0.7-μm pore size). Nano-POM samples were collected from the sieve-filtered water on pre-washed and pre-combusted 25 mm Whatman GF/D filters (2-μm pore size). Using the GF/D-filtered water, pico-POM samples were collected on pre-washed and pre-combusted 25 mm Whatman GF/F filters by further filtering. All filters were immediately frozen and stored at −20 °C until stable isotope analysis (SIA). For SIA, all filter samples were treated with HCl fumes to remove inorganic carbon and then dried. The three subsamples collected at each depth were then folded and wrapped with tin capsules.

We used a Bongo net (mesh size 330 μm; mouth diameter 60 cm) equipped with a flow meter (Rigosha No. 1313) for zooplankton sample collection (Figure 1). At the six stations of the East Sea, the Bongo net was obliquely towed from ~200 m depth to the surface at a ship speed of 2 knots. After sampling, we immediately preserved all zooplankton samples using 99% ethanol.

### 2.2. Zooplankton Community Structure and Stable Isotope Analysis

After sampling, zooplankton samples were divided using the Motoda zooplankton splitter until the number of copepods was over 600 individuals in each sample. The copepod species were then identified and counted using a dissecting microscope (Zeiss, Stemi 305). The rest of the samples collected at the two stations were used for SIA (Figure 1). We calculated the carbon mass of copepods in each station based on the equation [30] as follows: Log C = 3.07 log L − 8.37, where C and L are carbon mass (μg C m$^{-3}$) and prosome length (μm) of copepods, respectively.

For SIA, we sorted 16 copepods, namely *Calanus sinicus*, *Clausocalanus pergens*, *Corycaeus affinis*, *Ctenocalanus vanus*, *Eucalanus bungii*, *Mesocalanus tenuicornis*, *Metridia pacifica*, *Microcalanus pygmaeus*, *Neocalanus cristatus*, *N. plumchrus*, *Oithona atlantica*, *Paracalanus parvus* s.l., *Pseudocalanus minutus*, *P newmani*, *Scolecithricella minor*, and *Triconia conifera*, from the bulk samples. We measured stable carbon and nitrogen isotope ratios of females with the exception of *N. plumchrus* in the south region and *N. cristatus* in the north region, of which individuals of copepodid V stage (CV) were analyzed. All copepods were analyzed as individuals or in pooled samples of up to ~50 individuals. Five or eleven subsamples of each species were dried at 60 °C for 24 h using a drying oven and then packed in tin capsules. Syväranta et al. [31] found that the standard deviation of the stable carbon and nitrogen

isotope ratios in ethanol-preserved copepods was wider than that in frozen samples but not significantly. Moreover, Im and Suh (unpublished data) found no significant difference in the carbon and nitrogen stable isotope ratios between frozen and ethanol-preserved samples of *E. bungii* (t-test, $\delta^{13}$C: t = 1.49, *p* > 0.05, n = 6; $\delta^{15}$N: t = 0.36, *p* > 0.05, n = 6). Therefore, we assumed the potential effect of ethanol preservation on the carbon and nitrogen stable isotope ratios of copepods.

All filter and zooplankton samples were oxidized at high temperature (1020 °C) in a Costech Elemental Analyzer (ECS 4010), and the resultant $CO_2$ and $N_2$ were analyzed for stable isotope ratio using elemental analysis-isotope ratio mass spectrometry (EA-IRMS, Thermo Scientific Conflo IV) at the University of Alaska, Fairbanks. Each stable isotope abundance was expressed in $\delta$ notation according to the following expression: $\delta X = [R_{sample}/R_{standard} - 1] \times 1000$, where X is $^{13}$C or $^{15}$N and $R_{sample}$ and $R_{standard}$ are the corresponding ratio $^{13}$C/$^{12}$C or $^{15}$N/$^{14}$N between the sample and standard. Vienna PeeDee Belemnite and atmospheric nitrogen (air) were used as standard materials for $^{13}$C and $^{15}$N, respectively. Measurement precision was approximately 0.1 and 0.2‰ for $\delta^{13}$C and $\delta^{15}$N, respectively. Copepod samples were not defatted before SIA because of the resulting change of bias in $\delta^{15}$N after lipid extraction [32,33]. We adjusted the stable carbon isotope ratio of each copepod for lipids using the mass balance model for zooplankton with reference to Equation (5) of [34] as follows: $\delta^{13}C_{ex} = \delta^{13}C_{bulk} + 6.3\left(\frac{C:N_{bulk}-4.2}{C:N_{bulk}}\right)$, where $\delta^{13}C_{ex}$ is the predicted stable carbon isotope ratio of the defatted copepod sample, $\delta^{13}C_{bulk}$ and $C:N_{bulk}$ are the stable carbon isotope ratio, and C:N is the ratio of each copepod sample, respectively.

### 2.3. Statistical Analysis

We tested the significance of differences among the stable carbon and nitrogen stable isotope ratios of vertical profiles of pico- to micro-POM and copepods using non-parametric ANOVA (Kruskall–Wallis). Multiple comparisons of the stable carbon and nitrogen stable isotope ratios of vertical profiles of pico- to micro-POM and copepods were tested using the Conover post hoc test [30]. The statistical significance level was set at 5%. Statistical tests were run by BrightStat [35]. To estimate the contributions of the vertical profiles of pico- to micro-POM to the resource utilization of copepods and the contributions of the vertical profiles of micro-POM to the diets of carnivorous copepods, including *C. affins* and *M. pygmaeus*, we used the SIAR isotope mixing model [36]. In the SIAR, the contributions of food sources to consumer diets are expressed as 95%, 75%, and 50% credibility intervals of the estimates. The SIAR incorporates variations in stable isotopic values from both vertical profiles of pico- to micro-POM and copepods in addition to trophic enrichment factors [37]. The trophic enrichment factors of carbon and nitrogen for copepods were 0.3 ($\pm$1.14) and 2.2 ($\pm$2.05), respectively [38].

## 3. Results

### 3.1. Temperature, Salinity, and Chlorophyll A Concentration

Figure 2 shows temperature, salinity, and fluorescence data in this study region. Water mass properties showed remarkable differences between the south and north regions of the SPF. The temperature at the surface layer ranged from 13.5 to 15.15 °C in the south region of the SPF, which was higher than the 5.17 to 8.49 °C range in the north region. The average temperature within the ~200 m depth in the south and north regions of the SPF ranged from 9.95 to 11.12 °C and from 2.13 to 1.55 °C, respectively. In both regions of the SPF, chlorophyll *a* concentration increased from the surface to SCM layers and then decreased; the concentration in the surface layer was higher in the south region of the SPF (0.71 $\pm$ 0.32) than in the north region (0.11 $\pm$ 0.04).

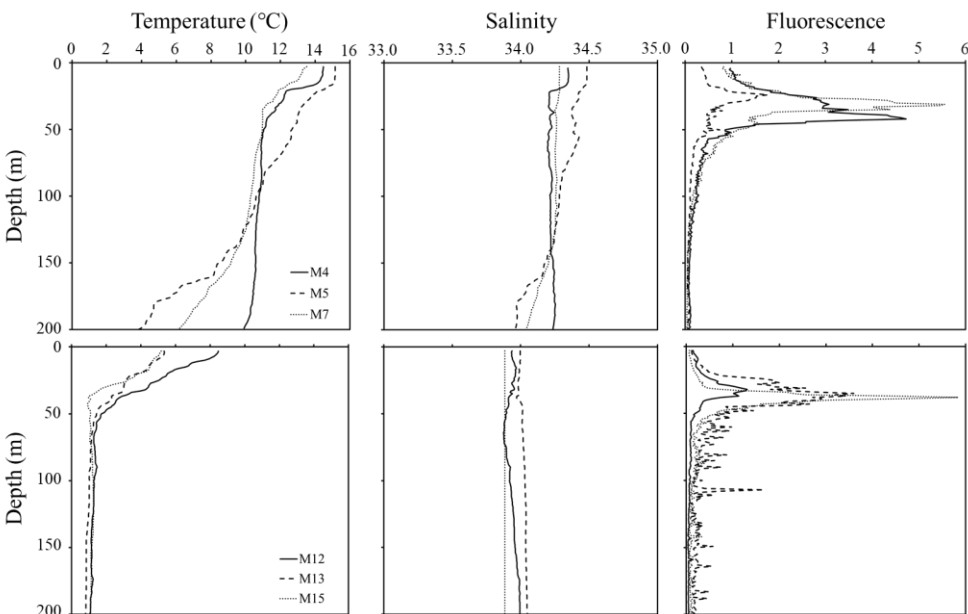

**Figure 2.** Vertical structures of temperature, salinity, and fluorescence at the southern (**A**) and northern stations (**B**) of the subpolar front.

### 3.2. Copepod Abundance and Carbon Masses across the SPF

In total, 16 copepod species were identified, and their abundances and carbon masses ranged from 64.04 to 322.31 ind. m$^{-3}$ and from 9.02 to 26.89 mg C m$^{-3}$, respectively (Figure 3; Table S1). The numerical abundance and species richness of copepods were higher in the south region of the SPF, while the carbon mass showed the reverse trend. Among copepods, *M. pacifica* and *N. plumchrus* occurred in the entire region and comprised 13–39% and 3–35% of the numerical abundance and 13–61% and 3–61% of the carbon mass in copepods, respectively. Moreover, *C. sinicus* and *M. tenuicornis* were represented only in the south region while *E. bungii* and *N. cristatus* occurred mainly in the north region.

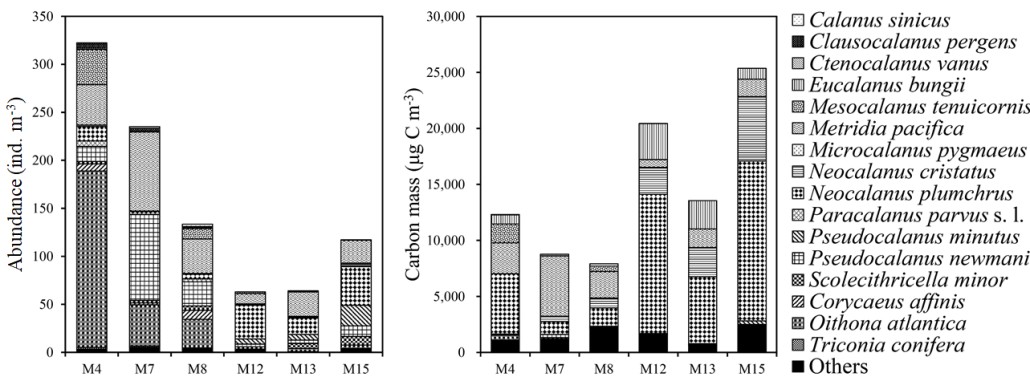

**Figure 3.** Copepod abundances (individuals (ind.) m$^{-3}$) and carbon masses in the south (M4–M7) and north regions (M12–M15) of the subpolar front.

### 3.3. Vertical Profiles of $\delta^{13}$C and $\delta^{15}$N Values of Pico- to Micro-POMs

Figure 4 and Tables S2 and S3 show the vertical profiles of stable carbon and nitrogen isotope ratios of pico- to micro-POM in the south and north regions of the SPF. In both regions of the SPF, the $\delta^{13}$C and $\delta^{15}$N values among pico- to micro-POM within ~200 -m depth layers were significantly different (ANOVA; $\delta^{13}$C: df = 11, $\chi^2$ = 38.57, $p < 0.001$; $\delta^{15}$N: df = 11, $\chi^2$ = 38.34, $p < 0.001$).

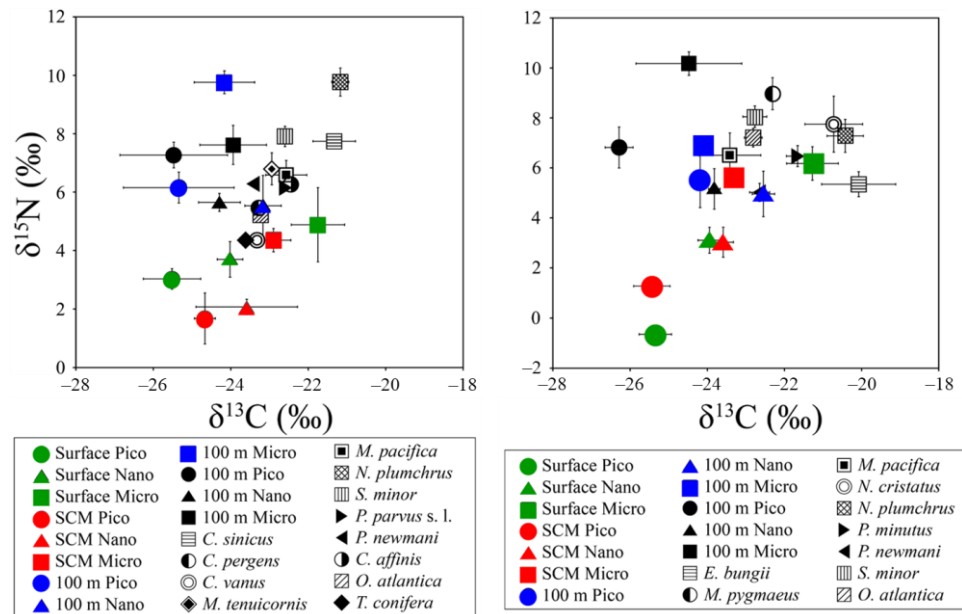

**Figure 4.** Mean (±SD) δ¹³C versus δ¹⁵N values of pico- to micro-particulate organic matter (POM) within ~200 m depth layers and for copepods in the south (**A**) and north (**B**) regions of the subpolar front.

In contrast, post hoc comparisons show that the δ¹³C and δ¹⁵N values of pico- and nano-POM between the SCM and 100 m depth layers in the south region (Table S4) and in those of nano-POM between the surface and SCM layers in the north region were not significantly different (Table S5). In both regions, the average δ¹³C values of pico- and nano-POM among were not significantly different with depth whereas those of micro-POM decreased gradually. In contrast, δ¹⁵N values of pico- and micro-POM showed a gradual increase with depth.

### 3.4. Carbon and Nitrogen Stable Isotope Ratios of Copepods

The average δ¹³C and δ¹⁵N values of copepods in the south region ranged from −23.60 ± 0.01 to −21.17 ± 0.22‰ and from 4.36 ± 0.37 to 9.77 ± 0.48‰, respectively (Figure 4; Table S2) whereas those in the north region ranged from −23.11 to −20.25‰ and from 5.02 to 9.30‰, respectively (Figure 4; Table S2). According to the Conover post hoc test, δ¹³C and δ¹⁵N values between *C. pergens* and *O. atlantica* and among *M. pacifica*, *P. newmani*, and *C. affinis* in the south region were not significantly different (Table S4). In the north region, δ¹³C and δ¹⁵N values were not significantly different between *E. bungii* and *N. plumchrus*, between *M. pacifica* and *P. newmani*, and between *S. minor* and *O. atlantica*. The δ¹³C values of copepods increased gradually with body size in both regions (Table S5). The increase in δ¹⁵N values of copepods with increasing body size occurred in the south region while those in the north region showed the reverse trend (Figure 5).

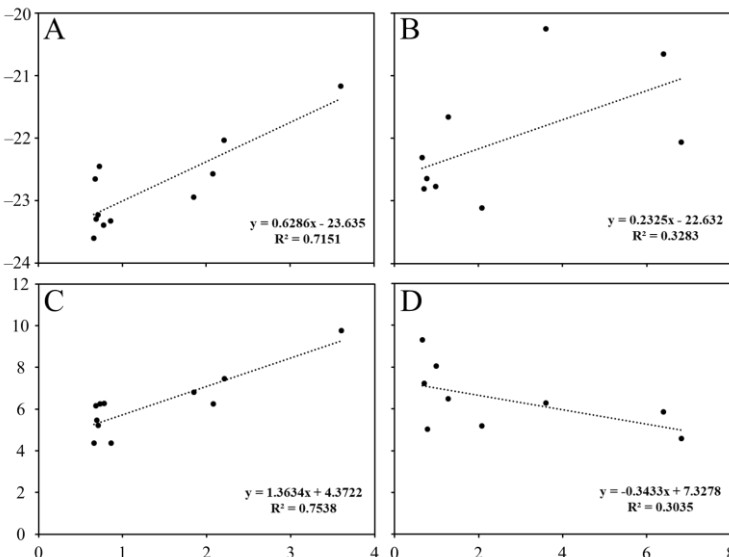

**Figure 5.** Mean prosome length versus $\delta^{13}$C and $\delta^{15}$N values of copepods in the south (**A,B**) and north regions (**C,D**) of the subpolar front.

The SIAR model results showed that resource utilization in most copepods occurred in the surface and SCM layers (Figures 6 and 7). Relatively high contributions of POM in the 100 m and 200 m depth layers to diets occurred in *N. plumchrus* in the south region of the SPF (Figure 6) and in *M. pygmaeus*, *O. atlantica*, and *S. minor* in the north region (Figure 7).

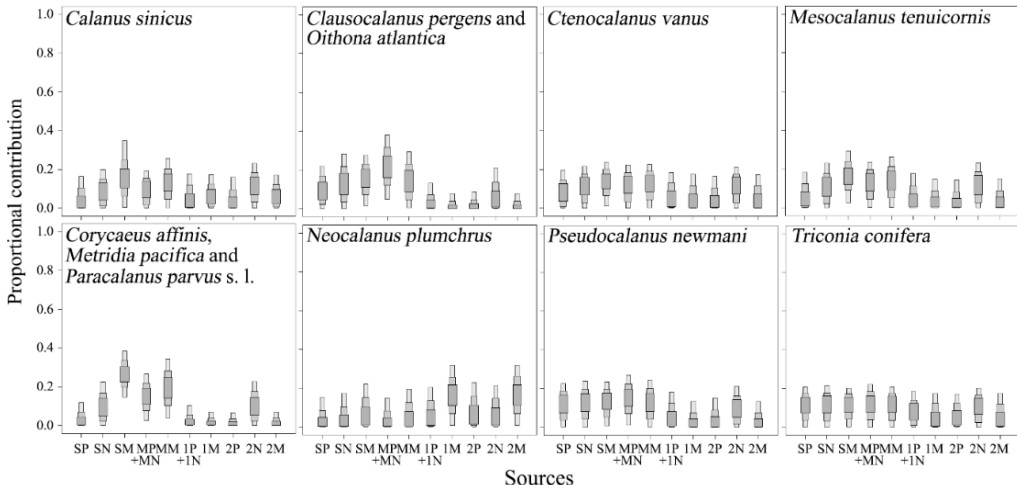

**Figure 6.** The stable isotope analysis in R mixing model results show contributions of pico-, nano-, and micro-particulate organic matter (POM) at the surface (0 m, 10 m), SCM, 100 m, and 200 m depth layers to the diets of copepods in the south region of the subpolar front using $\delta^{13}$C and $\delta^{15}$N values. The shaded boxes (dark to light) show 50%, 75%, and 95% credibility intervals for mean estimates. SP, surface layer pico-POM; SN, surface layer nano-POM; SM, surface layer micro-POM; MP, SCM layer pico-POM; MN, SCM layer nano-POM; MM, SCM micro-POM; 1P, 100 m depth layer pico-POM; 1N, 100 m depth layer nano-POM; 1M, 100 m depth layer micro-POM; 2P, 200 m depth layer pico-POM; 2N, 200 m depth layer nano-POM; 2M, 200 m depth layer micro-POM.

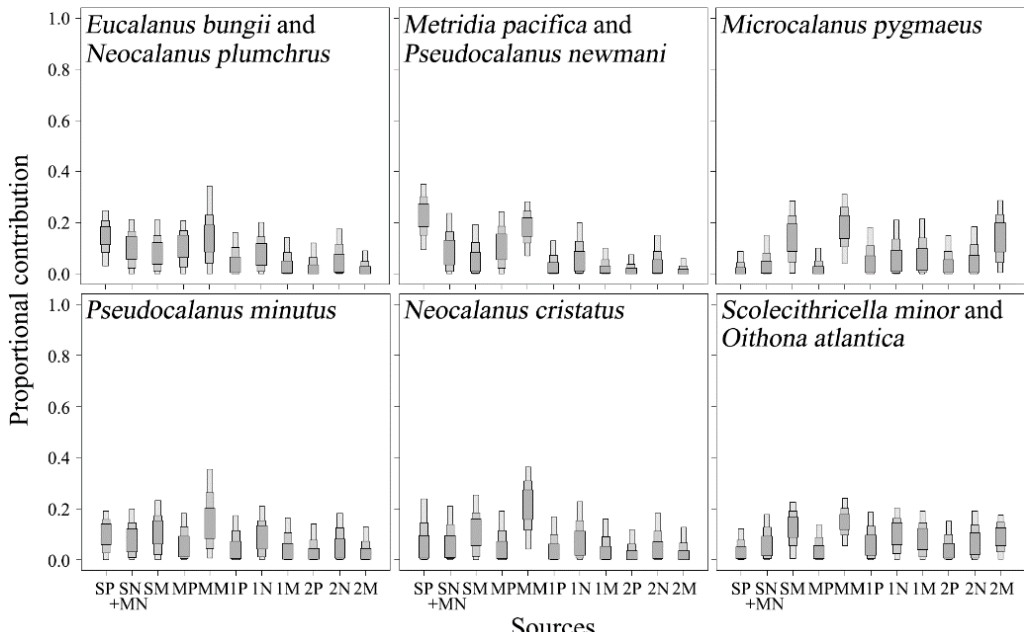

**Figure 7.** The stable isotope analysis in R mixing model results show the contributions of pico-, nano-, and micro-particulate organic matter (POM) at the surface (0 m, 10 m), SCM, 100 m, and 200 m depth layers to the diets of copepods in the north region of the subpolar front using $\delta^{13}C$ and $\delta^{15}N$ values. The shaded boxes (dark to light) show 50%, 75%, and 95% credibility intervals for mean estimates. SP, surface layer pico-POM; SN, surface layer nano-POM; SM, surface layer micro-POM; MP, SCM layer pico-POM; MN, SCM layer nano-POM; MM, SCM micro-POM; 1P, 100 m depth layer pico-POM; 1N, 100 m depth layer nano-POM; 1M, 100 m depth layer micro-POM; 2P, 200 m depth layer pico-POM; 2N, 200 m depth layer nano-POM; 2M, 200 m depth layer micro-POM.

Based on the trophic relationship between micro-POM and small copepods and the diets of *C. affinis* in the south region and *M. pygmaeus* in the north region, the SIAR results showed that *C. affinis* fed mainly on small copepods rather than on micro-POM in the surface and SCM layers while *M. pygmaeus* fed on micro-POM in the 100 m and 200 m depth layers as well as on *O. atlantica* and *S. minor* (Figure 8).

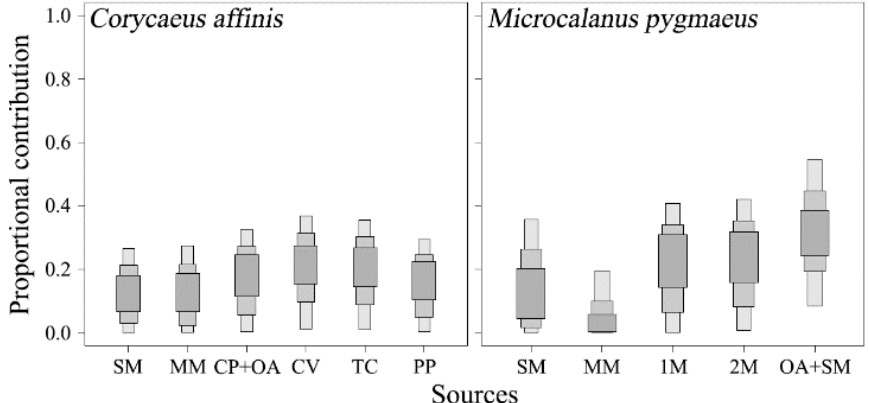

**Figure 8.** The stable isotope analysis in R mixing model results show the contributions of micro-particulate organic matter (POM) at the surface (0 m, 10 m), subsurface chlorophyll, 100 m, and 200 m depth layers, and small copepods to the diets of *Corycaeus affinis* in the south region of the subpolar front and *Microcalanus pygmaeus* in the north region using $\delta^{13}C$ and $\delta^{15}N$ values. The shaded boxes (dark to light) show 50%, 75%, and 95% credibility intervals for mean estimates. SM, surface layer micro-POM; MM, SCM micro-POM; 1M, 100 m depth layer micro-POM; CP, *Clausocalanus pergens*; CV, *Ctenocalanus vanus*; OA, *Oithona atlantica*; SM, *Scolecithricella minor*; TC, *Triconia conifera*.

## 4. Discussion

We estimated the trophic relationship between 16 copepods and the vertical profiles of pico- to micro-POM within the epipelagic region in the East Sea. Based on the functional traits reported by [14], *C. sinicus*, *C. vanus*, *E. bungii*, *M. pacifica*, *M. tenuicornis*, *N. cristatus*, *N. plumchrus*, and *P. parvus* s. l. are grouped under the filter-feeding copepods. Previous studies have suggested that the size-dependent morphology of the filter-feeding copepods influences the prey-size preference and inter- and intraspecific differences in dietary components [29,39]. The SIAR results showed that the contribution of micro-POM to the diets of filter-feeding copepods increased gradually with increasing body size (Figures 6 and 7). Moreover, most of the filter-feeding copepods fed mainly on the surface- and SCM-layer POM, with the exception of *N. plumchrus* in the south region, which mainly consumed micro-POM at the 100 m and 200 m depth layers (Figures 6 and 7). Thus, it is likely that the size-related morphology and vertical segregation of feeding habitats among copepods can promote coexistence within the same water column.

Among filter-feeding copepods, the feeding migration ranges of *N. plumchrus* differed across the SPF (Figures 6 and 7). Because the $\delta^{13}$C and $\delta^{15}$N values of *N. plumchrus* in the north region were not significantly different between CV ($\delta^{13}$C, $-20.43 \pm 0.47$‰; $\delta^{15}$N, $7.28 \pm 0.66$‰) and female adults, the difference in its feeding migration between the south and north regions might be affected by environmental factors, such as temperature and food supply [40,41]. *N. plumchrus* is a sub-arctic copepod and may be transported into the south region of the SPF through the southward North Korea Cold Current [42]. The much higher temperature at the surface and SCM layers in the south region than in the north region might limit the distribution range of *N. plumchrus* to below the 100 m depth (Figure 2). Moreover, chlorophyll *a* concentration in the surface layer was much higher in the south region than in the north region (Figure 2), indicating relatively high primary productivity at the surface layer in the south region [43]. The phytoplankton community is the initial source of detrital POM and plays a key role in controlling the recycling and export of organic materials into and out of the euphotic zone [44]. In south regions of the SPF, the high primary production at the surface layer in spring might supply organic materials below the euphotic zone [45]. Therefore, sinking particles resulting from high primary production in the surface layer might provide enough energy for *N. plumchrus* population in the south region of the SPF.

Small calanoid copepods, including *C. pergens*, *M. pygmaeus*, *P. minutus*, and *P. newmani* were grouped into small sac-spawning herbivores based on functional traits [27]. However, our isotopic results showed that *C. pergens*, *P. minutus*, and *P. newmani* mainly fed on the surface and SCM layer POM, while the contribution of the 200 m depth layer micro-POM to the resource utilization of *M. pygmaeus* was relatively high (Figures 6 and 7). Schnack-Schiel et al. [46] suggested that the gnathobase morphology of *M. pygmaeus* with very long and pointed teeth similar to those of euchaetid copepods showed the predatory behavior of feeding on zooplankton rather than on phytoplankton. Previous studies on the vertical distribution of *M. pygmaeus* showed that no clear seasonally vertical migration and sexes of adults occurred below the thermocline throughout the day [44]. Based on the contribution of micro-POM from the surface to the 200 m depth layer and small copepods to its resource utilization, the SIAR results showed that the relatively high $\delta^{15}$N values of *M. pygmaeus* could be derived from the consumption of higher trophic status materials, such as deep-layer POM or small copepods rather than surface and SCM layer micro-POM (Figure 8). However, we could not conclude whether *M. pygmaues* fed mainly on small copepods because the contributions of small copepods to its diet were not remarkably higher than that of micro-POM in the 100 m and 200 m depth layers (Figure 8). More studies on the carnivorous feeding of *M. pygmaeus* should be conducted.

*S. minor* is considered to be a typical detritus feeder with specialized setae on the maxillae and maxillipeds to detect the chemical signals of prey [47,48]. In the north region, *S. minor* showed relatively high $\delta^{15}$N values (Figure 4) and fed on micro-POM from the surface to 200 m depth layers (Figure 7). These results indicated that *S. minor* showed an

extensive range of feeding migration. However, previous studies on the vertical migration showed that *S. minor* was distributed below the thermocline throughout the day [49,50]. These are not consistent with the SIAR results analyzed in this study. In contrast, based on the trophic relationship between POM below the thermocline and *S. minor*, the SIAR results showed that *S. minor* fed on nano- and micro-POM in the 100 m and 200 m depth layers rather than POM in the SCM layer (data not shown), thus supporting that the relatively high $\delta^{15}$N values of *S. minor* might result from the consumption of POM in the deep layers with higher trophic status [49,50].

Small cyclopoid copepods, such as corycaeid, oithonid, and oncaeid, commonly occur in various ecosystems, although their feeding behavior remains unknown compared with calanoids [51]. Despite the similarity of raptorial mouth parts, previous studies on gut content analysis indicated differences in feeding ecology among cyclopoid copepods [52–54]. In this study, we found significant differences in $\delta^{13}$C and $\delta^{15}$N values among *C. affinis*, *O. atlantica*, and *T. conifera* in the south region of the SPF (Figure 4; Table S2), which supports the previous suggestions. Among these copepods, *C. affinis* is grouped into small carnivores [14], although its $\delta^{13}$C and $\delta^{15}$N values were not significantly different from those of *M. pacifica* and *P. parvus* s. l. (Table S2). However, the SIAR results based on the contribution of micro-POM and small copepods to the diet of *C. affinis* revealed that despite the relatively low $\delta^{15}$N values, *C. affinis* mainly fed on small copepods, such as *C. vanus* and *T. conifera* rather than micro-POM in the surface and SCM layers (Figure 8). Therefore, we suggest that *C. affinis* plays an important role in the linkage between small copepods and higher trophic organisms in the south region. In contrast, oncaeid copepods usually exhibit a wide vertical distribution and are known as a consumer of epipelagic POM associated with larvacean houses and small particles in aggregates or individual [55,56]. We found relatively depleted $\delta^{15}$N values of *T. conifera* (Figure 3), indicating herbivorous or omnivorous feeding rather than carnivorous feeding. The SIAR results showed that the contributions of pico- to micro-POM in the surface and SCM layers were relatively high (Figure 5). Thus, *T. conifera* might feed on pico- to micro-POM in the surface and SCM layers in aggregates via raptorial mouth parts [57].

*Oithona* spp. show trophic flexibility and trophic independence along environmental conditions, and it is difficult to estimate a particular ecological function [14]. In this study, $\delta^{13}$C and $\delta^{15}$N values of *O. atlantica* were not significantly different from those of *C. pergens* in the south region, which mainly fed on POM in the surface and SCM layers (Figure 6) and from those of *S. minor* in the north region, which mainly fed on micro-POMs in the 100 m and 200 m depth layers. There are two possible explanations with regard to the difference in the feeding migration pattern of *O. atlantica* between the south and north regions. First, the abundance and carbon masses of *O. atlantica* are closely associated with the higher chlorophyll *a* concentration in the surface layer in the south region of the SPF than in the north region (Figures 2 and 3). This indicates that high primary production in the surface and SCM layers might support the higher abundance of *O. atlantica* in the south region of the SPF than in the north region. The other explanation is potential predators within the epipelagic zone in the north region. Large copepods, including *E. bungii* and *N. cristatus*, mainly occurred in the north region of the SPF (Figure 3), and the SIAR results showed that they mainly fed on POM in the surface and SCM layers (Figure 7). They are both potential competitors to feed on POM in the surrounding waters and/or predators to show potentially carnivorous feeding on small copepods [58]. Thus, higher $\delta^{15}$N values in small copepods than in large copepods in the north region of the SPF could result from the vertical segregation of feeding migration ranges (Figure 7). Therefore, we suggest that feeding migration and resource utilization of *O. atlantica* influence primary productivity and potential predators.

Our study provides information on the effect of the copepod community structure on energy flow in the pelagic ecosystem using functional ecology. Our isotopic results show interspecific and intraspecific differences in the feeding migration pattern and habitat uses among sympatric copepods. However, we could not demonstrate niche partitioning of

temporal dimension among copepods because zooplankton samples were collected by the Bongo net [59]. To estimate niche partitioning of temporal dimension among copepods, zooplankton samples should be collected using multiple opening/closing net systems during both day and night.

The resource utilization and feeding migration pattern of zooplankton can be influenced by spatio-temporal differences in water mass properties and prey concentration [41,60]. We found remarkable differences in the resource utilization and feeding migration pattern of zooplankton between the south and north regions of the SPF. In the south region of the SPF, higher species richness of copepods than in the north region occurred (Figure 3), while the body size of copepods that mainly fed on POM in the surface and SCM layers was considerably smaller than that of the copepods in the north region (Figures 6 and 7). Such results indicated a shorter food chain in the north region of the SPF than in the south region, which showed the transfer of more energy from primary producers to higher trophic levels in the north region [59].

Generally, the body size shows interspecific and intraspecific differences among crustaceans, including amphipods and copepods, and influences various biological rates and predator–prey interactions [39,60–62]. Previous studies showed that the trophic position of zooplankton gradually increased with the body size increase [63,64], indicating that zooplankton size structure reflects the trophic transfer through size-based predator–prey interactions [65,66]. However, we found that the stable carbon and nitrogen isotope ratios of copepods in the south region increased gradually with increasing body size while the stable nitrogen isotope ratios in the north region showed the reverse trend (Figure 4). As mentioned above, the relatively high $\delta^{15}$N values of small copepods in the north region were closely associated with the consumption of POM in the 100 m and 200 m depth layers or small copepods (Figure 7). This indicates that the size-related food web explanation can have a critical limitation on understanding energy flows in the pelagic ecosystem [14,67]. Therefore, we suggest that the species-based approach should be conducted to understand the effect of zooplankton community structure on energy flows in the pelagic ecosystem [68,69].

## 5. Conclusions

Interspecific and intraspecific differences in the resource utilization and feeding migration range among copepods in each region of the SPF could contribute to their coexistence within the same water column. The numerical abundance and species richness of copepods were higher in the south region of the SPF. Moreover, the body size of copepods that consumed POM in the surface and SCM layers was much smaller in the south region of the SPF than in the north region. Such results indicated the lower transfer efficiency of energy in the south region of the SPF derived from the longer food chain than in the north region. Therefore, northward shifts in the distribution of copepods under global warming might lead to a decrease in the transfer efficiency of energy in the epipelagic zone of the East Sea ecosystem. Moreover, we suggest that studies on the species-based approach may provide more useful information on the effect of changes in the copepod community structure on biogeochemical cycles in the pelagic ecosystem than size-related food web explanation.

**Supplementary Materials:** The following supporting information on can be downloaded at: https://www.mdpi.com/article/10.3390/w15030416/s1, Table S1: Numerical abundance (ind. m$^{-3}$) of copepods in the East Sea; Table S2: C/N ratio, stable carbon ($\delta^{13}$C) and nitrogen ($\delta^{15}$N) isotope ratios of food sources and copepods in the south region of the subpolar front of the East Sea; Table S3: C/N ratio, stable carbon ($\delta^{13}$C) and nitrogen ($\delta^{15}$N) isotope ratios of food sources and copepods in the north region of the subpolar front of the East Sea; Table S4: Conover post hoc test results for pair-wise differences in δ13C and δ15N values among pico- to micro-POMs within the ~200-m depth layer (a) and among 11 copepods (b), *Calanus sinicus, Clausocalanus pergens, Ctenocalanus vanus, Mesocalanus tenuiocrnis, Metridia pacifica, Neocalanus plumchrus, Paracalanus parvus* s. l., *Pseudocalanus newmani, Scolecitrhricella minor, Corycaeus affinis, Oithona atlantica,* and *Triconia conifera* in the south region of the subpolar front of the East Sea; Table S5: Conover post hoc test results for pair-wise differences in

δ13C and δ15N values among pico- to micro-POMs within the ~200-m depth layer (a) and among nine copepods (b), *Eucalanus bungii*, *Metridia pacifica*, *Microcalanus pygmaeus*, *Pseudocalanus minutus*, *P. newmani*, *Neocalanus cristatus*, *N. plumchrus*, *Scolecithricella minor*, and *Oithona atlantica* in the north region of the subpolar front of the East Sea.

**Author Contributions:** D.-H.I. and H.-L.S. designed the experiments and drafted the manuscript; D.-H.I. designed sample collection in the field, analyzed the community structure, and performed statistical analyses; D.-H.I. wrote the main manuscript text; H.-L.S. supervised all processes. All authors have read and agreed to the published version of the manuscript.

**Funding:** This study was a part of the project entitled "Deep Water Circulation and Material Cycling in the East Sea (20160040)", funded by the Ministry of Oceans and Fisheries (MOF), South Korea. This research was also supported by the National Institute of Fisheries Science, MOF (R2023014).

**Institutional Review Board Statement:** Not applicable.

**Informed Consent Statement:** Not applicable.

**Data Availability Statement:** The data presented in this study are available on request from the corresponding author.

**Acknowledgments:** We thank the captain and crew of R/V Lavrentyev for their assistance with sample collection.

**Conflicts of Interest:** The authors declare no conflict of interest.

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
