# Peer review of "Trophic Positions of Sympatric Copepods across the Subpolar Front of the East Sea during Spring: A Stable Isotope Approach"

_water, doi:10.3390/w15030416_

Round 1
Reviewer 1 Report (Previous Reviewer 1)
I approve the resubmitted manuscript. My compliments!
Author Response
Thank you for your constructive comments.
Reviewer 2 Report (Previous Reviewer 2)
Throughout text:
1-Referring to ranges is not necessary to repeat the unit, e.g. instead of “9.02 mg C m-3 to 26.89 mg 161 C m-3” use “9.02 to 26.89 mg 161 C m-3”, instead of “13% to 61%” use “13 to 61%”
2-The formal unit for abundance is ind m-3 not inds. m-3
3-Instead of “reported by [27]” use “reported by author name [27]”
4-Again, English need careful revision; tenses sometimes are not synchronized and there are sporadic syntax errors.
Conclusions: This part should be more concise. It should report original synthetic considerations emerged from the Discussion and highlight the most original findings of the work. Avoid repeating statements, already included in the discussion and not being conclusions.
L 405 – 407 Delete ”We investigated the species-specific …using SIA”.
L 409 – 411 Delete ”We found that the numerical …showed the reverse trend”
Or
L 409 - 413. Rephrase, keeping short comments on abundance, richness and body size.
Author Response
1-Referring to ranges is not necessary to repeat the unit, e.g. instead of “9.02 mg C m-3 to 26.89 mg 161 C m-3” use “9.02 to 26.89 mg 161 C m-3”, instead of “13% to 61%” use “13 to 61%”
==> We have revised the text as per your suggestion.
2-The formal unit for abundance is ind m-3 not inds. m-3
==> We have revised the text as per your suggestion.
3-Instead of “reported by [27]” use “reported by author name [27]”
==> We have revised the text as per your suggestion.
4-Again, English need careful revision; tenses sometimes are not synchronized and there are sporadic syntax errors.
==> We have attached the certification of English editing service.
Conclusions: This part should be more concise. It should report original synthetic considerations emerged from the Discussion and highlight the most original findings of the work. Avoid repeating statements, already included in the discussion and not being conclusions.
L 405 – 407 Delete ”We investigated the species-specific …using SIA”.
==> We have revised the text as per your suggestion.
L 409 – 411 Delete ”We found that the numerical …showed the reverse trend”
Or
L 409 - 413. Rephrase, keeping short comments on abundance, richness and body size.
==> Thank you for your comments. We have corrected the sentence as follows: “The numerical abundance and species richness of copepods were higher in the south region of the SPF. Moreover, the body size of copepods that consumed POMs in the surface and SCM layers was much smaller in the south region of the SPF than in the north region.”

Reviewer 3 Report (New Reviewer)
The authors presented an interesting study of the trophic position of copepods within epipelagic zone with the approach of stable isotope. Generally, the study is noteworthy in investigating spatially the variation among the species and size in East Sea. I have few comments with the present manuscript which I believe the authors can adequately address.
Comments follow:
L12: sympatric species of copepods maybe what you mean?
L12-13: please revise the sentence as all these prepositional comparison “among, within, across” in one sentence cause confusion.
L50: please add reference
L54-60: So here I suggest introducing in detail the mechanism of ectotherms range shifts due to warming. Generally, ectotherms shift their distribution to where they can have their optimal performance within their thermal limit. the extinction or absence in a water level occur when ambient temperature exceeds the upper thermal limits. See Shokri et al., 2022: Metabolic rate and climate change across latitudes: evidence of mass-dependent responses in aquatic amphipods. Rubalcaba, 2022: Oceanic vertical migrators in a warming world. Those works might be helpful to expand this part.
L117: the model of the microscope
L120: log(x), is it log10 or log2, please explain here.
L124-128: the species must be written in italic font, also please homogenously use the complete name of the species
L139-140: it is an assumption; thus I suggest rephrasing as “therefore, we assumed that using ethanol for preservation of copepods won’t affect the carbon and nitrogen stable isotope ratios”
L152: change “recommended” to “with referenced to Eq”
L176-177: The average has a value [± S.D], while you report a range. removing “average” from the sentence might be better to describe it. Otherwise explain why the average reported as the range.
L179: in this sentence and also earlier, I assumed they are significantly higher or lower; increased or decreased, right (p<0.05)?
Fig3: So M(x) must be the stations, to be easily read, please explain them in the figure legend, explained also which ones are belonging to North and which in South (M15,13,12:North), (M7…:South).
L214-215: in the figure 4, it is mean (±SD) while here it is written average, reported as a range, please make it clear why they are reported as the range.
L390-391: Also studies on other crustaceans showed that body size exert a major influence on prey preference, which support your finding. See Shokri et al, 2021:A new approach to assessing the space use behavior of macroinvertebrates by automated video tracking. Cozzoli et al, 2022: The size dependency of foraging behaviour: an empirical test performed on aquatic amphipods.
Author Response
The authors presented an interesting study of the trophic position of copepods within epipelagic zone with the approach of stable isotope. Generally, the study is noteworthy in investigating spatially the variation among the species and size in East Sea. I have few comments with the present manuscript which I believe the authors can adequately address.
Comments follow:
L12: sympatric species of copepods maybe what you mean?
==> We have used the word “sympatric” to convey the meaning of “coexisting.”
L12-13: please revise the sentence as all these prepositional comparison “among, within, across” in one sentence cause confusion.
==> We have revised the sentence as follows: “We investigated the trophic relationship between particulate organic matters (POMs) and sympatric copepods within the epipelagic zone (~200 m depth) in the East Sea during spring based on stable isotope analysis (SIA).”
L50: please add reference
==> We have revised the text as per your suggestion.
L54-60: So here I suggest introducing in detail the mechanism of ectotherms range shifts due to warming. Generally, ectotherms shift their distribution to where they can have their optimal performance within their thermal limit. the extinction or absence in a water level occur when ambient temperature exceeds the upper thermal limits.
See Shokri et al., 2022: Metabolic rate and climate change across latitudes: evidence of mass-dependent responses in aquatic amphipods.
Rubalcaba, 2022: Oceanic vertical migrators in a warming world.
Those works might be helpful to expand this part.
==> Thank you for your comments. We have added the sentence “Moreover, ectotherms shift their distribution pattern to where they can achieve optimal performance within their thermal limit.”
L117: the model of the microscope
==> We added the model of the microscope, "Zeiss, Stemi 305” in the sentence.
L120: log(x), is it log10 or log2, please explain here.
==> Log(x) here means “log10(x).”
L124-128: the species must be written in italic font, also please homogenously use the complete name of the species
==> We have revised the text as per your suggestion.
L139-140: it is an assumption; thus I suggest rephrasing as “therefore, we assumed that using ethanol for preservation of copepods won’t affect the carbon and nitrogen stable isotope ratios”
==> We have revised the text as per your suggestion.
L152: change “recommended” to “with referenced to Eq”
==> We have revised the text as per your suggestion.
L176-177: The average has a value [± S.D], while you report a range. removing “average” from the sentence might be better to describe it. Otherwise explain why the average reported as the range.
==> Thank you for your comment. We have revised the sentence to explain the average temperature within the ~200-m depth range at each station.
L179: in this sentence and also earlier, I assumed they are significantly higher or lower; increased or decreased, right (p<0.05)?
==> Thank you for your comment. Although we did not perform the statistical analysis because the number of samples was too small, Chl a concentration in the south region was significantly higher than that in the north region.
Fig3: So M(x) must be the stations, to be easily read, please explain them in the figure legend, explained also which ones are belonging to North and which in South (M15,13,12:North), (M7…:South).
==> We have revised the text as per your suggestion.
L214-215: in the figure 4, it is mean (±SD) while here it is written average, reported as a range, please make it clear why they are reported as the range.
==> We have revised the text as per your suggestion.
L390-391: Also studies on other crustaceans showed that body size exert a major influence on prey preference, which support your finding.
See Shokri et al, 2021:A new approach to assessing the space use behavior of macroinvertebrates by automated video tracking.
Cozzoli et al, 2022: The size dependency of foraging behaviour: an empirical test performed on aquatic amphipods.
==> Thank you for your comment. We have added the references in the sentence.

This manuscript is a resubmission of an earlier submission. The following is a list of the peer review reports and author responses from that submission.
Round 1
Reviewer 2 Report
water-1818363
Trophic positions of sympatric copepods across the subpolar front of the East Sea during spring: A stable isotope approach
Dong-Hoon Im and Hae-Lip Suh
This manuscript presents trophic relationships between particulate organic matters and 11 copepod species in the epipelagic zone across the subpolar front of the East Sea using stable isotope analysis. The work is well designed and has been carried out carefully. Although there is not any major problem, there are several weak points that should be improved. Especially in the discussion there are some unclear phrases and, in general, a lack of cohesiveness. English need careful revision; tenses sometimes are not synchronized and there are sporadic syntax errors.
Specific comments
INTRODUCTION
L52 - 55 "However, the difficult ... in the pelagic ecosystem." Not clear, to be rephrased
MATERIALS AND METHODS
L83 Replace "using methods of [21]" with "using methods described in [21]”
L91, 92 Replace "(individuals (inds.) m-3)" with “(ind. m-3)”. This is the standard expression of abundance.
L110 Replace "[26] found" with "Syvaranta et. al. [26] found"
L138, 139. Not clear, to be rephrased.
RESULTS
L 154, 155. To be checked if the difference is statistically significant. "much higher" is too subjective.
L 157. Figure 2; Correct temperature scales in A and B, use the same scale (as in the other two parameters).
L162 - 164 "Numerical abundance and species richness ….. the reverse trend". Provide a table including species richness.
L169 Figure 3; Redraw using different colors as, in black and white, is difficult to perceive the graph
L177 Figure 4; The same as above, use different colors
DISCUSSION
L263 Replace "reported by [24]" with "reported by Benedetti et. al. [24]"
L290 - 293. Not clear, explain further your speculation
L298, 299 Replace "[41] suggested" with "Schnack-Schiel and Mizdalski [41] suggested"
L322 - 324. Not clear. What do you mean with "higher trophic status" (maybe higher trophic level)? Please rephrase
L351 Instead of "explanation" use "hypothesis"
L246 -367 In general this paragraph is lacking cohesiveness. Please rewrite
L365 - 367 Unclear
L369 Delete "using the functional ecology"
L386 Replace "showed" with "supported"
L391 Replace "[60] found " with "Jung [60] found"
L394 Replace "[61] reported" with "Joo et. al. [61] reported"
L399 Replace "shows" with "is related with"
Reviewer 3 Report
Please see attached document
